# Control of Aflatoxigenic Molds by Antagonistic Microorganisms: Inhibitory Behaviors, Bioactive Compounds, Related Mechanisms, and Influencing Factors

**DOI:** 10.3390/toxins12010024

**Published:** 2020-01-01

**Authors:** Xianfeng Ren, Qi Zhang, Wen Zhang, Jin Mao, Peiwu Li

**Affiliations:** 1Oil Crops Research Institute of the Chinese Academy of Agricultural Sciences, Wuhan 430062, China; renxianfenga@163.com (X.R.); zhangwen@oilcrops.cn (W.Z.); maojin106@whu.edu.cn (J.M.); 2Key Laboratory of Biology and Genetic Improvement of Oil Crops, Ministry of Agriculture and Rural Affairs, Wuhan 430062, China; 3Key Laboratory of Detection for Mycotoxins, Ministry of Agriculture and Rural Affairs, Wuhan 430062, China; 4Laboratory of Risk Assessment for Oilseeds Products, Ministry of Agriculture and Rural Affairs, Wuhan 430062, China; 5Quality Inspection and Test Center for Oilseeds Products, Ministry of Agriculture and Rural Affairs, Wuhan 430062, China

**Keywords:** aflatoxin, biocontrol strategy, *Aspergillus*, prevention

## Abstract

Aflatoxin contamination has been causing great concern worldwide due to the major economic impact on crop production and their toxicological effects to human and animals. Contamination can occur in the field, during transportation, and also in storage. Post-harvest contamination usually derives from the pre-harvest infection of aflatoxigenic molds, especially aflatoxin-producing *Aspergilli* such as *Aspergillus flavus* and *A. parasiticus*. Many strategies preventing aflatoxigenic molds from entering food and feed chains have been reported, among which biological control is becoming one of the most praised strategies. The objective of this article is to review the biocontrol strategy for inhibiting the growth of and aflatoxin production by aflatoxigenic fungi. This review focuses on comparing inhibitory behaviors of different antagonistic microorganisms including various bacteria, fungi and yeasts. We also reviewed the bioactive compounds produced by microorganisms and the mechanisms leading to inhibition. The key factors influencing antifungal activities of antagonists are also discussed in this review.

## 1. Introduction

Aflatoxins are the most common contaminants occurring widely in oilseeds and grains. Aflatoxins B_1_, B_2_, G_1,_ and G_2_ are a group of potent hepatotoxic and carcinogenic secondary metabolites produced mainly by *Aspergillus* section *Flavi* spp. like *A. flavus* and *A. parasiticus* [1,2,3]. Aflatoxigenic molds can cause a decrease in production, a loss of nutritional value, and a diminution of market value of agricultural products, and also cause serious diseases like allergic reactions in humans and animals. Aflatoxin B_1_ (AFB_1_), the most toxic and commonly occurring one, has been classified as group Ι human carcinogen by the International Agency for Research on Cancer [4]. Aflatoxins M_1_ and M_2_, which are by-products of the above aflatoxins, may be found in dairy products from animals fed with contaminated feed and are closely related to the safety of dairy food.

Physical strategies such as field managements, physical separations, and moisture controls, and chemical strategies (e.g., using fungicides and chemical absorbents) have been applied to control aflatoxin and its producing molds. In most cases, the physical and chemical methods were inefficient, due to a nutritional loss of the processed foods, a difficulty in removing residues of the toxic compounds, or a development of resistant biotypes of pathogens. The biological control has been regarded as a more environmentally friendly and safer method [5,6], which was carried out generally at pre- and/or post-harvest. The post-harvest strategies focus mainly on the removal of aflatoxin [7,8,9,10,11,12]. However, once the agro-foods and feeds are contaminated, the contaminants such as aflatoxins can never be completely removed. Therefore, preventing aflatoxin production and fungal infection is the most efficient strategy.

In the past decades, many research studies have historically focused on the biocontrol of aflatoxigenic molds [13,14]. Three main modes of inhibitory actions are involved: antagonists grow rapidly to occupy ecological niche and compete for nutrients and/or living places, which leads to a displacement of pathogens; another involves inhibiting fungal growth, which leads to a reduction of fungal infection and colonization; the third is based on inhibiting aflatoxin biosynthesis. This review explores inhibitory behaviors, bioactive compounds, mechanisms of inhibitory actions, and factors influencing biological activities.

## 2. Antagonistic Microbes against Aflatoxigenic Strains

Various microorganisms including bacteria, fungi such as nontoxigenic *Aspergillus, Trichoderma* and *penicillium* spp., and yeast strains have been investigated as potential biocontrol agents against aflatoxigenic strains. As shown in Figure 1, the articles reporting bacterial antagonists were dominant (61%) compared with the articles reporting antagonistic fungi (27%) or yeasts (12%). Additionally, a comprehensive list of all microorganisms (approximately 50 different species) that have been well documented for their anti-aflatoxigenic potential is given in Table 1. Main characteristics and inhibitory behaviors of these antagonists are described as well.

### 2.1. Bacteria

#### 2.1.1. *Bacillus* spp.

*Bacillus* spp. are a multifunctional group of bacteria. As shown in Figure 1, 21% of research articles reported *Bacillus* spp., which were most widely assessed in controlling aflatoxigenic strains. Aflatoxin accumulation in potato dextrose broth was almost totally inhibited by *B. megaterium* [16]. *B. subtilis* was also able to inhibit *A. parasiticus* growth and aflatoxin production by a percentage up to 92% and 100%, respectively [15]. Thus, *B. megaterium* and *B. subtilis* showed the highest biocontrol activity, inhibiting the growth of as well as aflatoxin production by aflatoxigenic strains, while *B. amyloliquefaciens* was also able to reduce *A. parasiticus* growth as well as degrade aflatoxins B_1_, B_2_, G_1,_ and G_2_ after several days of co-cultivation [53,54]. González et al. [18] demonstrated that *B. mojavensis, B. cereus,* and *B. mycoides* isolated from soil had ability to significantly inhibit *A. parasiticus* growth. Isolates of *B. pumilus* were also demonstrated with ability to inhibit aflatoxin production [17]. As reviewed by Schallmey et al. [55], *Bacillus* spp. were intensively assessed as biological agents, probably because they grew rapidly, produced a wide range of antimicrobial compounds, and generally were recognized as safe species. 

#### 2.1.2. *Pseudomonas* spp.

It was found that *P. fluorescens* could reduce AFB_1_ production by *A. flavus* in peanut medium at a rate of 99.4% [20], as well as inhibit conidial germination of *A. flavus* by up to nearly 20% [56]. A known fact is that *Pseudomonas* is one of the most prevalent genera isolated from soil (plants rhizosphere or nonrhizosphere). Palumbo et al. [19] demonstrated that the chitinolytic *P. chlororaphis* strains isolated from maize fields and maize rhizospheres could completely inhibit *A. flavus* growth. Mannaa et al. [21] found that *P. protegens* strain AS15 isolated from rice grains also significantly inhibited aflatoxin production by and mycelial growth of *A. flavus* at rates of 82.9% and 68.3%, respectively. Several other *Pseudomonas* strains were also demonstrated with an ability to completely inhibit the growth of *A. flavus* in different media [34].

#### 2.1.3. *Lactobacillus* spp.

Lactic acid bacteria (LAB) are bacteria producing organic acids—mainly lactic acid—by carbohydrate fermentation. In food production, these bacteria are traditionally used to prevent spoilage and increase shelf life of foods. As shown in Table 1, *L. plantarum*, *L. rhamnosus*, *L. casei*, *L. fermentum*, *L. pentosus*, *L. paraplantarum,* and *L. delbrueckii subsp. Lactis* have been identified as biocontrol agents against aflatoxigenic fungi. Ahlberg et al. [57] demonstrated that LAB strains showed an ability to physically bind aflatoxins. In another study of Ahlberg et al. [26], 171 LAB strains were tested against *A. flavus*, and the species with the highest antifungal ability was identified as *L. plantarum*. The genus *Lactobacillus*, mainly the species *L. plantarum*, has been widely found to inhibit aflatoxigenic strains in various living environments [22,58,59,60,61]. LAB strains are mainly divided into four genera: *Lactobacillus, Lactococcus, Pediococcus,* and *Leuconostoc*. As was reported by Sangmanee and Hongpattarakere [22], the supernatant obtained from *L. plantarum* culturing broth could inhibit the mycelial growth and aflatoxin production of *A. flavus* by 100%. *L. casei*, *L. fermentum*, *L. reuteri,* and *L. acidophilus* were also proved to have an inhibitory effect higher than 80% on *Aspergillus niger*, *Penicillium* sp., and *Fusarium graminearum* [25]. Ahlberg et al. [26] demonstrated that *Lactobacillus* spp. with high or moderate anti-mycotoxigenic activities were identified as *L*. *pentosus*, *L. paraplantarum,* and *L. plantarum*. Species of *L. delbrueckii subsp. Lactis* were also found to completely inhibit aflatoxin G_2_ production and significantly control *A. parasiticus* growth [27].

#### 2.1.4. *Streptomyces* spp.

*Streptomyces* spp. are gram (+) filamentous bacteria that widely grow in soils and on plants. A *Streptomyces* strain isolated from peanuts was found to completely inhibit, directly or via secondary metabolites, mycelial growth and conidial germination of *A. flavus* [62]. *Streptomyces* strain ASBV-1 was found to be able to reduce the viability of *A. parasiticus* spores and subsequently, inhibit aflatoxin accumulation in peanut grains [63]. Verheecke et al. [64] reported that several soil-born *Streptomyces* isolates had a strong bioactivity against aflatoxin B_1_ and B_2_ production by *A. flavus*. Several other *Streptomyces* species (Table 1) have been evaluated as bioactive agents providing an antagonistic activity against aflatoxigenic isolates. Shakeel et al. [28] demonstrated that culture filtrates and crude extracts of *S. yanglinensis* could completely inhibit mycelial growth of *A. flavus*. Studies demonstrated that *S. anulatus* [29], *S. alboflavus* [30], and *S. roseolus* [31] also exerted an effective antifungal activity toward aflatoxigenic strains and other common agricultural crops pathogens. 

#### 2.1.5. Other Bacteria Species

*Serratia marcescens* strain JPP1 isolated from peanut hulls is an endophytic bacterium which lives inside the plant tissue and does not cause visible morphological changes. Strain JPP1 exhibited remarkable inhibitory effects on aflatoxin production (rate >98%) and mycelial growth (rate >95%) of *A. parasiticus* [32]. *Stenotrophomonas* sp., a soil bacterium, could produce inhibitors against aflatoxin production, but without affecting fungal growth [33]. *Nannocystis exedens*, a myxobacterium commonly found in soil, had a potential to control the growth of *A. flavus* and *A. parasiticus* by lysing pathogens’ colony [35]. Palumbo et al. [34] isolated 171 bacteria from California almond orchard samples; apart from the familiar genera *Bacillus* and *Pseudomonas*, *Burkholderia cepacia*, *B. pyrrocinia*, *Delftia acidovorans*, *D. acidovorans,* and *Ralstonia paucula* were also demonstrated with potential activity against *A. flavus* growth. *Achromobacter xylosoxidans,* a gram-negative and catalase-positive bacterium, is already known to have wide biological control abilities [65]. Yan et al. [36] demonstrated that *A. xylosoxidans* could produce inhibitory substances remarkably inhibiting *A. flavus* and *A. parasiticus* growth.

According to the current studies, antagonistic bacteria were actually highly effective on aflatoxigenic strains in vitro. However, their colonization in soil and on crops has not been evaluated under field conditions. Due to genetic and environmental differences, it is not easy to bring the bacterial cells to the *Aspergilli* infection sites. This may be the reason why most of the anti-aflatoxigenic studies are performed only in vitro, and no bacterial agents are already commercialized.

### 2.2. Fungi

#### 2.2.1. Nontoxigenic *Aspergillus* spp.

*A. flavus* are variable in respect to aflatoxin-producing ability, and were described into S (small) and L (large) strains on the basis of sclerotial morphological types [66]. On average, S strains produce higher levels of AFB_1_ with less variation in aflatoxin production [67,68]; L strains are more variable in aflatoxin production and even include nonproducers entirely lacking the ability to produce aflatoxins [69]. Currently, introduction of nontoxigenic *A. flavus* into fields is the most promising strategy for preventing pre-harvest aflatoxin contamination. The use of nontoxigenic *A. flavus* to competitively exclude aflatoxigenic strains was first introduced by Cotty and Bayman [70]. As shown in Figure 1, there have been many studies subsequently focusing on nontoxigenic *Aspergillus*. Prevention of aflatoxin accumulation by inoculation with nontoxigenic *A. flavus* CT3 and K49 was assessed in a 4year field study [37], with results indicating that the reduction percentages of aflatoxin on southern US corns were 65–94%. Alaniz Zanon et al. [38] also found that the nontoxigenic *A. flavus* had a higher biocontrol efficacy against aflatoxin accumulation (inhibition rate = 78–90%) in a two-year study in northern Argentina. In addition, nontoxigenic *A. flavus* isolates were demonstrated with an ability to reduce aflatoxin contamination of maize by a rate higher than 80% in Kenya [71]. Importantly, nontoxingenic *A. flavus* strains, AF36 (NRRL 18543) and Afla-Guard^®^ (NRRL 21882), have been commercialized for use in groundnut and maize production, respectively, in USA. This biocontrol approach has also been proved to be effective on peanut [72,73], cottonseed [69], and corn [37,74] under field conditions. An application of nontoxigenic *A. parasiticus* in the field was also able to reduce aflatoxin contamination in storage [39]. Nontoxigenic *A. niger* strain FS10, isolated from fermented soybean, could not only significantly inhibit *A. flavus* growth, but also inhibit AFB_1_ production (rate = 94.5%) [40,75]. *A. oryzae,* the nontoxigenic domesticated ecotype of *A*. *flavus*, is used as a “Generally Recognized As Safe (GRAS)” microorganism for food fermentation [76,77]. Alshannaq et al. demonstrated that co-inoculation with *A. oryzae* and *A. flavus* on peanuts with a ratio of 1:100 could effectively inhibit AFB_1_ production [41]. The species of *A. clavatus* could secrete ribonuclease [78], while Skouri-Gargouri and Gargouri revealed that *A. clavatus* could inhibit the growth of several plant pathogens such as *Fusariuym oxysporum* and *Aspergillus niger* due to the secretion of an antifungal peptide [42].

#### 2.2.2. *Trichoderma* spp.

*Trichoderma* spp. comprise a large number of rhizocompetent filamentous strains in soils and root ecosystems. Their potential as fungal biocontrol agents against plant pathogenic fungi has been known for a long time [79]. The majority of *Trichoderma* isolates used industrially for biological control belong to the species *T. harzianum*, including strains T22 and T39 [80,81]. *Trichoderma* species can not only control crop diseases, but also exert beneficial effects on root growth and enhance crop productivity [82]. There have been several *Trichoderma* species reported showing varying degrees of control of aflatoxigenic strains since last century [79]. *T. harzianum* and *T. viride* were proved to be highly antagonistic and inhibit mycelial growth and aflatoxin production of *A. flavus* by a rate higher than 80% [43]. Evaluation of *Trichoderma* spp. for biocontrol of pre-harvest seed infection by *A. flavus* in groundnut was performed by Anjaiah et al. [44], with results indicating that in greenhouse and field experiments, the treatment of seeds with *Trichoderma* spp. including *T. harzianum*, *T. longibrachiatum*, *T. viride,* and *T. auroviride* reduced *A. flavus* populations (as cfu) by a percentage higher than 50%. A known fact is that *Trichoderma* species are historically a group of the most studied beneficial filamentous fungi. Sarrocco and Vannacci [14] gave a list of commercial bio-pesticides containing 14 different *Trichoderma* strains that belong to *T. harzianum*, *T. atroviride*, *T. virens*, *T. asperellum*, *T. gamsii,* and *T. polysporum*; however, these species have not been investigated as commercialized products to biocontrol aflatoxigenic molds.

#### 2.2.3. *Penicillium* spp.

Several species of *Penicillium* are able to grow rapidly in the presence of toxigenic strains [83]. The strain RP42C of *P. chrysogenum* was reported as antifungal-protein producer with a biological activity against the growth of aflatoxigenic strains on dry-cured ham [45,84]. *P. chrysogenum*, a fungal starter culture for mold-fermented foods production, is related to *P. nalgiovense*. Nielsen et al. [85] demonstrated that *P. nalgiovense* showed a higher inhibitory effect on the growth of the common fungal pathogens. Additionally, Geisen [46] demonstrated that *P. nalgiovense* had a greater inhibition on the secondary metabolites production of fungal strains. As fungal starter cultures, the antifungal activity of *Penicillium* species would play an important role in the safety of mold-fermented food.

### 2.3. Yeast Strains

Due to the ability to consume lactic acid in the presence of oxygen, yeast strains have been regarded as deteriorating agents for a long time. Yeast strains are also popular in household because of the ability of leavening dough. Marine yeast *Debaryomyces hansenii* BCS003 strain can decrease mycelial growth by almost 98% in a radial inhibition assay against *Aspergillus* strains [47], while native *D. hansenii* strains were also demonstrated with a significantly antagonistic activity on the growth rate and aflatoxin production of *A. parasiticus* in meat products [48]. *Saccharomyces cerevisiae* RC008 and RC016 are strains demonstrated with the ability of inhibiting the growth of and AFB_1_ production by *A. parasiticus* under different regimes of water activities, pH values, temperatures, and oxygen availabilities [49]. As shown in Table 1, *Kluyveromyces*, *Pichiaanomala,* and *Candida maltosa* isolates were also demonstrated to have an impact on mycelial growth, conidial germination, or aflatoxin production when interacting with aflatoxigenic *Aspergillus* strains [48,50,52,86,87,88].

### 2.4. A Conclusion of Antagonistic Microbes

Many bacteria agents have been demonstrated with an ability to inhibit aflatoxigenic molds; however, none of bacterial agents has been commercialized. At current research status, only nontoxigenic *A. flavus* NRRL 18543 and NRRL 21882 have been commercialized and applied in fields [72], and *Trichoderma* species are just showing a high potential to be commercialized for the future use [14]. For yeast strains, however, we need further studies to look for strains with high efficacy.

## 3. Inhibitory Compounds Produced by Different Antagonistic Microbes

Secondary metabolites produced by various microorganisms are high-value natural products, many of which exhibit significant pharmacological properties. The inhibitory compounds discussed here are secondary metabolites with powerful bioactive properties in biological control of aflatoxin-producing fungi. Based on the results obtained in vitro experiments, inhibitory compounds produced by various antagonistic microorganisms and their bioactivities against aflatoxigenic molds are listed in Table 2. These compounds are divided into four different types of substances, including micromolecular organics, organic acids, antibiotics, and enzymes (Figure 2). The following paragraphs describe the producers, anti-aflatoxigenic activities, and main characteristics of these compounds in more detail.

### 3.1. Antibiotics and Proteases Produced by Bacillus *spp.*

*Bacillus* species generally have characteristics to produce antimicrobial substances, mainly including lipopeptides, protease antibiotics, and bacteriocins [100]. These structurally diverse compounds exhibit a wide range of antimicrobial activity [101], especially the lipopeptides secreted by *Bacillus* presenting antifungal activity [102]. 

*Bacillus* strains isolated from aquatic environments were evaluated for their antifungal effect on *A. flavus* and *A. carbonarius*, producers of AFB_1_ and ochratoxin A, respectively [89]. Results showed that the lipopeptides (iturin A and surfactin isomers in extracts) produced by *Bacillus* sp. P1 strain exhibited high anti-*Aspergillus* activities on mycelial growth, conidial germination, and AFB_1_ and ochratoxin A production. Veras et al. [89] also analyzed the extracts from supernatants and cell pellets, and results indicated that lipopeptides were extracted mainly from cell-free supernatants. González Pereyra et al. [18] also demonstrated that lipopeptides, the extracellular compounds produced by soil *Bacillus* strains, were able to almost completely inhibit *A. parasiticus* growth and AFB_1_ production. In another report [103], mutants of *B. subtilis* obtained after varying doses of gamma irradiation could significantly inhibit *A. flavus* growth and aflatoxin production in pistachio nuts compared with the parental strain, because lipopeptides production of mutants increased. Additionally, Farzaneh et al. [104] reported that cell-free supernatants from *B. subtilis* had a significant effect on *A. flavus* spores viability, and the mass spectrometric analysis revealed that surfactin and fengycin were responsible for the biocontrol activity. These studies indicated that fengycin, surfactin, and iturin families of lipopeptides produced by *Bacillus* species were the dominant compounds potentially reducing *Aspergillus* spp. growth or aflatoxins production and, generally, these compounds were obtained from cell-free supernatants. Bacillomycin D, a lipopeptide substance produced by *B. subtilis,* was also demonstrated with abilities of significantly affecting mycelial growth, sporulation, and destabilizing the cell wall and cell membrane of *A. flavus* [90]. 

Proteases, especially alkaline proteases, are the well-known products of *Bacillus* strains. *B. subtilis* and *B. amyloliquefaciens* were able to inhibit *A. parasiticus* growth and showed a good proteolytic activity [15]. Additionally, three peptides of L-Asp-L-Orn (D1O), L-Asp-L-Asn (D1N), and L-Asp-L-Asp-L-Asn (D2N) produced by *B. megaterium* could significantly inhibit the growth of *A. flavus* [91]. Another study reported that unknown volatiles produced by *B. megaterium* could inhibit aflatoxin production, mycelial growth, and conidial germination of *A. flavus* in rice grains [105]. 

Overall, we can conclude from these studies that extracellular compounds of *Bacillus* species were able to inhibit aflatoxigenic molds. The compounds, especially lipopeptides and proteases may be the main effective antifungal factors inhibiting aflatoxin production, sporulation, and conidial germination and reducing mycelial growth.

### 3.2. Chitinolytic Enzyme Produced by Pseudomonas *spp.*

Akocak et al. [56] demonstrated that the chitinolytic enzyme produced by *P. fluorescens* could reduce the growth of *A. flavus* by inducing the morphological changes on conidial germination and mycelial growth. As reviewed by D’Aes et al. [106], biosurfactants such as cyclic lipopeptide and rhamnolipid produced by *Pseudomonas* spp. were involved in important functions of biocontrol. Phenazines produced by *Pseudomonas* strains were also major determinants controlling several plant pathogens [107]. However, biological activities of bio-surfactants and phenazines against aflatoxigenic strains have not been investigated. Therefore, speeding up the identification of bioactive compounds could potentially enhance application values of *Pseudomonas* species.

### 3.3. Organic Acids and Peptides Produced by Lactobacillus *spp.*

As revealed by Russo et al. [61], Lactobacillus spp. have broad antifungal activities because of the high production of lactic acid. Apart from lactic acid, phenyllactic acid (PLA), hydroxyphenyllactic acid (OH-PLA), and indole lactic acid (ILA) were also found to strongly inhibit aflatoxin-producing fungi [25,92]. Additionally, the antifungal compounds secreted by L. plantarum were investigated against the growth of and aflatoxin production by A. flavus and A. parasiticus, with results indicating that the antifungal compounds obtained from the cell-free supernatant, apart from lactic acid, majorly were 2-butyl-4-hexyloctahydro-1H-indene, oleic acid, palmitic acid, linoleic acid, and 2,4-di-tertbutylphenol [22]. 

Apart from organic acids, inhibitory peptides produced by *L. plantarum* were also demonstrated to be effective against *A. flavus* and *A. parasiticus* [59,60], while organic acids were dominant, probably associated with their low pH values [61].

### 3.4. Micromolecular Organics, Organic Acids, and Enzymes Produced by Streptomyces *spp.*

*Streptomyces* spp., known to produce over 7500 bioactive compounds including anticancer agents, vitamins, and antibiotic compounds, have a better tolerance to water stress [28]. They usually do not secrete toxic residues that may contaminate environments because of their natural origin. 2-methylisoborneol, the volatile organic compound generated by *S. alboflavus*, was proved to have an ability of inhibiting *A. flavus*, *Fusarium moniliforme,* and *Penicillum citrinum* in vitro [30]. Aflastatin A, extracted from mycelial cake of *Streptomyces* sp., was a strong inhibitor of aflatoxin production [93]. Dimethyl trisulfide and Benzenamine, the small molecular organic compounds generated by *S. alboflavus*, played an important role in controlling aflatoxin production and *A. flavus* growth [95]. Dimethyl disulfide, the micromolecular volatile organic identified from the volatiles of *S. alboflavus*, was proved to act as an antagonistic substance against some plant pathogens in vitro [30]. Dioctatin A, an organic acid, produced by *Streptomyces* spp., was found to strongly inhibit aflatoxin production and conidiation of *A. parasiticus* [94]. The thermostable endochitinase purified from *Streptomyces* sp. [96] and the chitinase (Chi242) obtained from the culture supernatant of *S. anulatus* [29] have been found to inhibit the mycelial growth of *A. parasiticus* and *A. niger*, respectively. From these studies, we are able to see out that inhibitory compounds produced by *Streptomyces* spp. were highly species-specific. As Manivasagan et al. [108] reviewed, *Actinomycetes*, especially *Streptomyces* spp., have a tremendous potential to produce various secondary bioactive metabolites. In this case, *Streptomyces* species definitely have a great potential to be used for the biocontrol of aflatoxigenic fungi.

### 3.5. Micromolecular Organics and Enzymes Produced by Yeast Strains

Yeast strains are increasingly targeted for the production of bioactive substances, especially the budding yeast species *Saccharomyces cerevisiae,* which has been proven to be a powerful microorganism for heterologous expression of biosynthetic pathways [109]. The biocontrol activity of *Pichia anomala* WRL-076 was attributed to the production of 2-phenylethanol, which was the major volatile compound affecting the growth, aflatoxin production, and gene expression of *A. flavus* [51]. Studies also demonstrated that isoamyl acetate and isoamyl alcohol produced by *Candida maltosa* were able to inhibit the conidial germination of *Aspergillus brasiliensis* [52]. 4-Hydroxyphenethyl alcohol, 4,4-Dimethyloxazole, and 1,2-Benzenedicarboxylic acid dioctyl ester in the supernatant extracts of *Saccharomyces cerevisiae* provided the antifungal activity against aflatoxigenic growth and aflatoxins biosynthesis [97]. Tayel et al. [98] demonstrated that *Pichia anomala* was able to produce β-1,3-glucanase and exo-chitinase, which were suggested as a mode of antifungal action leading to cause hyphal lysis of *A. flavus*.

### 3.6. Protease and Extracellular Enzymes Produced by Trichoderma *spp*.

Regarding to *Trichoderma* species, only a few inhibitory compounds that play roles in their antagonistic interactions with aflatoxingenic fungi were reported. Deng et al. [99] demonstrated that the aspartic protease P6281 secreted by *T. harzianum* could efficiently inhibit the conidial germination and the growth of *A. flavus*. Mostafaet al. [43] demonstrated that *T. harzianum* and *T. viride* showed a high antagonism and inhibited aflatoxins production of *A. flavus* by 90%, which were explained partially by the liberation of extracellular enzymes and the production of inhibitory volatile compounds.

### 3.7. Inhibitory Compounds Produced by the Other Microorganisms

Apart from the inhibitory compounds described in the above sections, chitinase produced by *Serratia marcescens* was able to efficiently degrade fungal cell walls [32]. The antifungal protein PgAFP produced by *Penicillium chrysogenum* could inhibit the growth of toxigenic molds [84]. Antifungal peptide produced by *Aspergillus clavatus* was thermostable and exhibited a strong inhibitory activity against mycelial growth of several plant pathogenic fungi [42]. Cyclo (L-Leucyl-L-Prolyl) produced by *Achromobacter xylosoxidans* was able to inhibit the growth of *A. parasiticus*, and it also remarkably repressed the transcription of the aflatoxin-biosynthesis related gene *aflR* [36].

### 3.8. A Conclusion of Inhibitory Compounds

Approximately 30 different compounds have been found to be bioactive against aflatoxigenic fungi. According to these studies, we identified three deficiencies in the research field that need improvement: (1) the variety of inhibitory compounds is still limited; and (2) all of the inhibitory compounds were tested only in vitro, in which case, it is difficult to relate with the real antagonistic efficacy in vivo because of the diversity of microbes in soils, differences of soil temperature, humidity, and pH, and the genetic and metabolic complexity of biocontrol antagonists; and (3) most, even all of the studies focused only on inhibitory efficiency, however, studies such as the resistance in *Aspergillus* and interactions among inhibitory compound, pathogen, antagonist, and environment were scarce. These deficiencies could be mirrored by the example of *Trichoderma* spp. The antagonistic *Trichoderma* strains have the ability to produce various compounds with antibiotic activity [81]. However, few antibiotic compounds have been identified from *Trichoderma* spp. for the biocontrol of aflatoxigenic molds. Although *Trichoderma* species play an important role in biocontrol of plant diseases, frequently enhance root growth, and induce systemic resistance responses of plants [82], the interaction among aflatoxigenic fungi, *Trichoderma*, soil, and plants has not been elucidated yet.

## 4. Mechanisms of Inhibitory Actions

### 4.1. Inhibitory Mechanisms by Antagonistic Bacteria

For antagonistic bacteria, their bioactive metabolites play a major role in controlling *Aspergillus* spp. growth and subsequent aflatoxin production. Inhibitory mechanisms by antagonistic bacteria mainly include (1) lysis of hyphae or spores by destablizing structure and composition of cell wall; (2) probably affecting intracellular activities of mitochondria, cytoplasmic membrane, and nucleus; and (3) down-regulating expression of aflatoxin-synthesis related genes. Illustrations were made as follows: chitinolytic enzymes produced by *P. fluorescens* reduced the growth of *A. flavus* by altering the germination pattern of spores [56]; the cell-free supernatant of *L. plantarum* caused morphological changes in seven-day-old *A. flavus* and *A. parasiticus*, because of severe damage to the mitochondria and nucleus, formation of the membrane-bound vesicles, and degeneration of the cytoplasmic membrane [22]; and dioctatin A produced by *Streptomyces* decreased expression of *aflR* and *brlA* (encoding a condition-specific transcription factor) and significantly inhibited the production of norsolorinic acid and sterigmatocystin that were precursors for aflatoxin synthesis [94].

### 4.2. Inhibitory Mechanisms by Nontoxigenic Aspergillus *spp.*

Fungal invasion, colonization, and competition between aflatoxigenic and atoxigenic strains of *A. flavus* have been studied [70,110]. Regarding nontoxigenic *Aspergillus* spp. as antagonist, two mechanisms are dominant: (1) toxigenic strains are physically excluded by the displacement of nontoxigenic strains during infection; and (2) nontoxigenic strains competed for nutrients that were required for aflatoxin biosynthesis. However, as Ehrlich [111] reviewed, there were a lot of challenges to using nontoxigenic *Aspergillus* species. Primarily, due in part to inherent diversity of *Aspergillus* species and genetic complexity, genetic mutations may happen in nontoxigenic *Aspergillus* spp., which potentially leads atoxigenic strains to mutate to aflatoxigenic strains; therefore, from a long-term security, nontoxigenic *Aspergillus* strains were also suggested to be cautiously used [112,113,114].

### 4.3. Inhibitory Mechanisms by Antagonistic Yeasts

How did the antagonistic yeasts act as biological agents to control aflatoxigenic growth and aflatoxin production? That the yeast strain *Pichia anomala* could efficiently inhibit the growth of and aflatoxin production by *A. flavus* can be attributed to the production of 2-phenylethanol, which led to remarkable effects on conidial germination and expression of genes necessary for aflatoxin biosynthesis [51], and the production of chitinase and glucanase, which led to hyphal lysis and deterioration [98]. That *Debaryomyces hansenii* was able to control *A. flavus* growth was attributed to the production of extracellular compounds and the competition for nutrients and spaces [47]. For *Saccharomyces cerevisiae*, the production of exochitinase and extracellular secondary metabolites could explain its mode of action for antifungal activity on the growth of *A. flavus* [97]. Therefore, for yeast strains, possible mechanisms of the inhibitory actions may involve two: (1) inhibiting aflatoxigenic growth by the production of extracellular enzymes and metabolites which lead to spores and hyphal deterioration, and (2) inhibiting aflatoxin production by down-regulating expression of aflatoxin biosynthesis genes.

### 4.4. Inhibitory Mechanisms by Antagonistic Trichoderma Strains

The antagonistic properties of *Trichoderma* strains are based on the activation of multiple physical and chemical mechanisms. The physical mechanisms included faster growth speed to compete for nutrients and living space, and mycoparasitism mediated by physical contact. Common interactions between antagonistic fungi and pathogens were divided into the following types [79]: 1 = antagonist overgrowing pathogen and pathogen stopped;1/2 = antagonist overgrowing pathogen but pathogen still growing; 2/1 = pathogen overgrowing antagonist but antagonist still growing; 2 = pathogen overgrowing antagonist and antagonist stopped;3 = mutual inhibition ≤2mm distance;4 = extremely mutual inhibition >4mm distance.

Calistru et al. [115] discovered only three interaction types between *Trichoderma* and *A. flavus,* namely antagonist overgrowing pathogen with growth inhibition of pathogen, pathogen overgrowing antagonist with growth inhibition of antagonist, and mutual inhibition. By a scanning electron microscopical investigation, Calistru et al. [115] revealed that mycoparasitism is not the mechanism of the inhibitory interaction between *A. flavus* and *Trichoderma* spp. (*T. harzianum* and *T. viride*). Conversely, Mostafa et al. [43] drew a conclusion that the aggressive behavior towards *A. flavus* by *T. harzianum* was explained by mycoparasitism.

The chemical mechanisms were also involved in producing cell walllytic enzymes and inducing the plant’s defense system to resist pathogens [116]. The production of extracellular enzymes was responsible for the inhibitory effect of *T. viride* on toxigenic *A. flavus* [43]. *T. harzianum* actively attached to the toxigenic *Aspergillus* species followed by enzymatic lysis of the mycelial filaments [117]. Such, mechanisms of the inhibitory actions against the growth of *A. flavus* by *T. harzianum* are strains-specific and mainly include (1) faster growth speed to compete for nutrients and living space, (2) mycoparasitism, and (3) the production of extracellular enzymes, deteriorating aflatoxigenic mycelia. However, the research on the mechanism of inhibitory effects on aflatoxin production is still at initial stage.

### 4.5. A Conclusion of Mechanisms

According to all of the above studies, we listed four main mechanisms of inhibitory actions (Figure 3): (1) Physically competing for living spaces and nutrients, (2) destabilizing cell wall structure, (3) affecting intracellular activities of mitochondria, nucleus, and cytoplasmic membrane, and (4) down-regulating expression of aflatoxin-synthesis related genes. Importantly, inhibitory actions are most likely determined by a combination of different mechanisms, not by only one.We also listed some genes that have been analyzed under the treatment of different biocontrol agents (Figure 4).

## 5. Factors Influencing Antifungal Activities

It is a well-known fact that the growth rate of and aflatoxin production by aflatoxigenic strains were strongly influenced by environments, cultural conditions, and nutritional factors. The combined effects of incubation time, temperature, water activity (aw), and CO_2_ on the growth of and aflatoxin production by *A. flavus* were studied [119]. Nutritional sources were also demonstrated to have a significant influence on fungal growth and mycotoxin production [120,121]. Additionally, the expression of aflatoxin-synthesis related genes were demonstrated to be highly in relation to changes in water activity and temperature levels [122,123,124]. Similarly, antifungal activities of various biocontrol microbes were also related to these biotic and abiotic factors. Examples are described here below.

### 5.1. pH Value

Studies showed that the bioactivity of *L. plantarum* was pH-dependent. The low pH was responsible for the highlighted bioactivity of *L. plantarum* against aflatoxin-producing strains [22,61]. Gerez et al. [25] demonstrated that the antifungal activity of some *Lactobacillus* strains was lost after the neutralization treatment because the acidic nature of the antifungal metabolites was destroyed. In addition, *Saccharomyces cerevisiae* RC008 and RC016 showed a great antagonistic activity at pH 4, where strains can highly decrease the growth rate of *A. parasiticus* [49]. Conversely, the bacterium *Bacillus pumilus* grew very slightly at pH 4, where it showed the lowest anti-aflatoxigenic activity only with 38% inhibition of aflatoxin production [17]. These studies indicated that the best pH value for different antagonists against aflatoxigenic molds is remarkably species-dependent.

### 5.2. Temperature and Water Activity

Culturing temperature and water activity (aw) are also key factors. The maximum activity of protease P6281 produced by *T. harzianum* was observed at 40 °C [99]. The appropriate conditions for the growth of *Kluyveromyces* spp. were 60 min of incubation at 45 °C and 0.95 aw [125], while Penna and Etcheverry [86] demonstrated that *Kluyveromyces* isolates could impact both *A. flavus* growth and AFB_1_ accumulation at a wide range of water activities (0.93–0.99). La Penna et al. [50] found that several *Kluyveromyces* isolates showed anti-aflatoxigenic activity and inhibitory activity on aflatoxin production at all water activities tested. A notable finding was that the yeast strains of *Debaryomyces hansenii* could stimulate aflatoxins production by *A. parasiticus* at water activity of 0.99, whereas significantly reduce aflatoxins production at 0.92 aw [48]. Therefore, temperature and water activity are also important factors influencing antifungal efficiency of antagonists.

### 5.3. Other Factors such as Incubation Time, Culturing Medium, and Mutagenesis

Furthermore, incubation time is also a key factor affecting the production of anti-aflatoxigenic metabolites. Munimbazi and Bullerman [17] gave an evident proving that the greatest inhibitory activity arose up after 3 and 4 days incubation of *Bacillus pumilus*, and aflatoxin production was completely inhibited in supernatant obtained only from 3 and 4 day old bacterium. Whipps [79] demonstrated that different media appeared to be related to antifungal behaviors. Afsharmanesh et al. [103] found that a random mutagenesis of *Bacillus subtilis* could significantly inhibit *A. flavus* growth and aflatoxin production compared with the parental strain. This shows that mutant study can potentially improve biocontrol activity in inhibiting aflatoxigenic strains.

As shown in Figure 5, incubation conditions such as growing period, temperature, water activity, pH values, and nutritional sources could not only influence pathogens’, but also antagonists’ growth and/or metabolism. Therefore, dynamic growing conditions should be taken into account in performing strategies to biocontrol aflatoxigenic molds and eliminate aflatoxin risk by aflatoxigenic fungi.

## 6. Perspective and Conclusions

The biocontrol strategy for preventing aflatoxigenic fungi has been discussed in this review. It is clear that some microbes, including various bacteria, nontoxigenic *Aspergillus*, *Trichoderma,* and yeasts have shown potentials to biocontrol aflatoxigenic molds. The inhibitory compounds that have potential biocontrol effects on aflatoxigenic strains, together with mechanisms and influencing factors of the bioactive actions are also reviewed. The current research status is still not very optimistic, because there are still many aspects needing urgent improvements. The above reviewed research works do, however, suggest that deeper practical works must be conducted to identify effective and environmental biocontrol agents, substantially to reach an advanced stage of application and commercialization. Additionally, a comprehensive and systematic study, covering inhibitory behaviors, mechanisms, factors, and pathogen–antagonist–plant interactions, is also urgently needed.

## Figures and Tables

**Figure 1 toxins-12-00024-f001:**
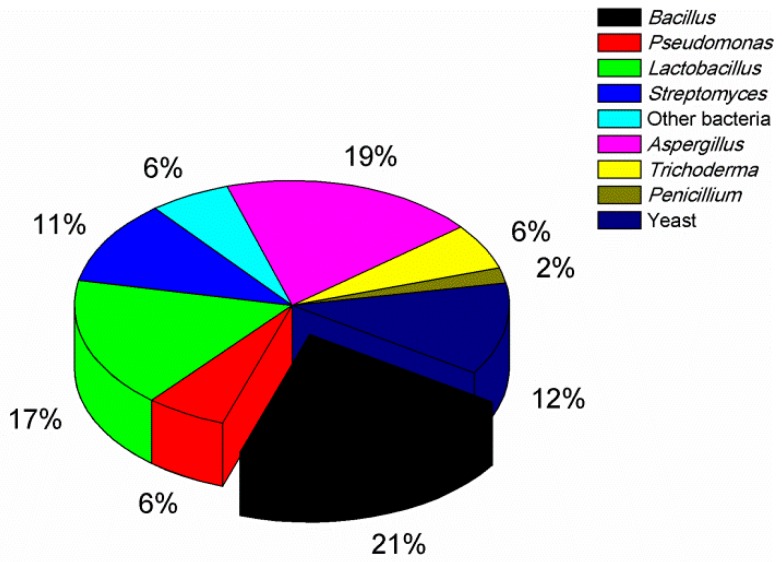
Percentages of research articles related to different antagonists of aflatoxigenic fungi. We searched for research articles on the topic of “biocontrol of aflatoxigenic fungi” on Web of Science (http://www.webofknowledge.com). Related research articles account for approximately 150, and each slice of the pie represents a percentage of the articles reporting each sort of microorganisms.

**Figure 2 toxins-12-00024-f002:**
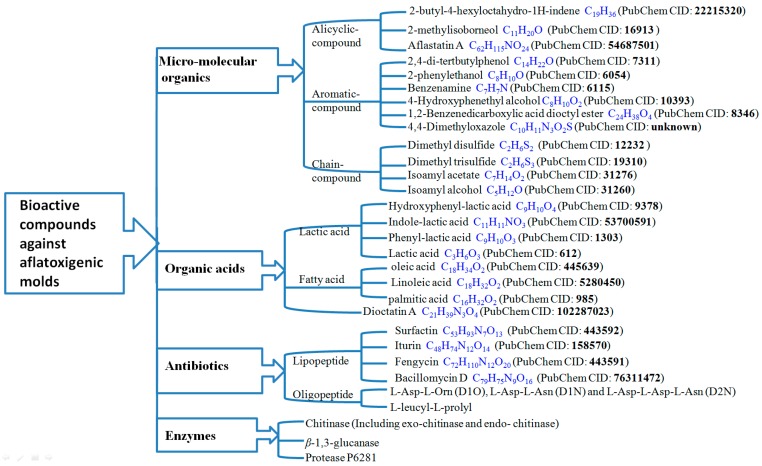
Bioactive compounds produced by microorganisms with antagonistic activities against aflatoxigenic molds. These compounds were divided into four different types of substances (micromolecular organics, organic acids, antibiotics, and enzymes). PubChem CID is listed at the end of each molecule. Details such as structures, molecular formula, and chemical and physical properties could be obtained in the following link: https://pubchem.ncbi.nlm.nih.gov/.

**Figure 3 toxins-12-00024-f003:**
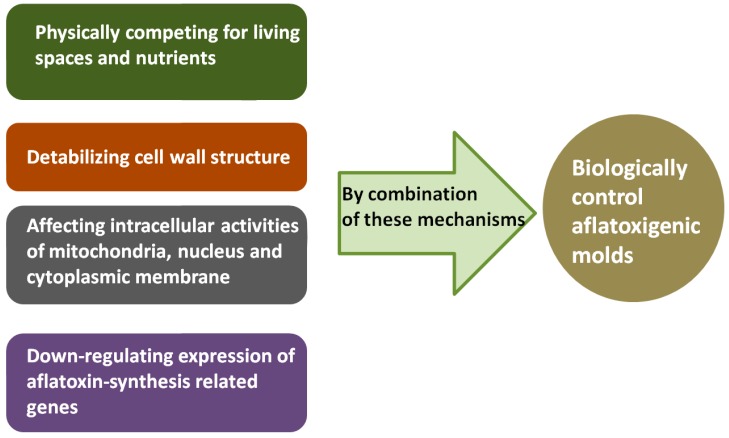
Mechanisms of inhibitory actions by antagonistic microorganisms against aflatoxigenic molds. For an inhibitory action, one of the four mechanisms may be dominant, but not the only one; inhibitory actions are most likely determined by a combination of different mechanisms.

**Figure 4 toxins-12-00024-f004:**
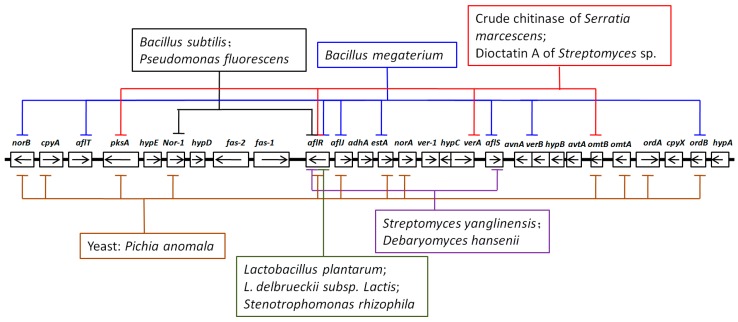
The genes down-regulated by different biocontrol agents. Different biocontrol agents acted on different aflatoxin synthesis genes which were demonstrated to be down-regulated. For example, *Bacillus subtliis* and *Pseudomonas fluorescens* could down-regulate the expressions of *Nor-1* and *aflR*. The clustered genes in aflatoxin biosynthetic pathway were plotted according to reports of Yu et al. [118].

**Figure 5 toxins-12-00024-f005:**
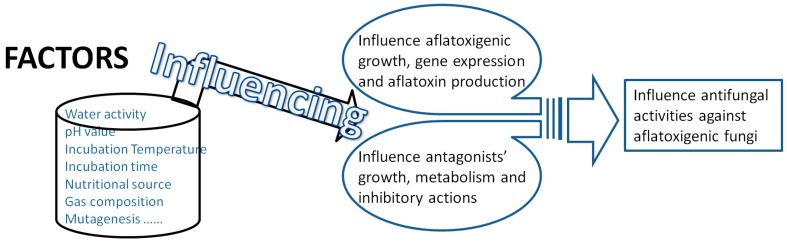
Key factors influencing antifungal activities against aflatoxigenic fungi. These key factors have influences on both baflatoxigenic and antagonists’ growth and metabolisms. As a result, the combination of these factors is playing an important role in biocontrol efficacy.

**Table 1 toxins-12-00024-t001:** Species evaluated for their activities on aflatoxigenic molds.

Microorganism	Genus	Specie	Activity	References
Bacteria	*Bacillus*	*B. subtilis*, *B. amyloliquefaciens*, *B. megaterium*, *B. mojavensis*, *B. cereus*, *B. pumilus*	Inhibit the growth of *A. flavus* and *A. parasiticus* Inhibit aflatoxin production	[15] [16] [17,18]
	*Pseudomonas*	*P. fluorescens*, *P. chlororaphis*, *P. protegens*	Inhibit *A. flavus* growth in grains	[19,20,21]
	*Lactobacillus*	*L. plantarum*, *L. rhamnosus*, *L. casei*, *L. fermentum*, *L. pentosus*, *L. paraplantarum*, *L. delbrueckii subsp. Lactis*	Bind aflatoxin M1 Inhibit aflatoxin production Inhibit fungal growth	[22,23,24] [25,26] [27]
	*Streptomyces*	*S. yanglinensis*, *S. anulatus*, *S. alboflavus*, *S. roseolus*	Inhibit *A. flavus* growth Inhibit *A. flavus* growth	[28,29] [30,31]
	Other bacteria	*Serratia marcescens*, *Stenotrophomonas* sp., *Ralstonia paucula*, *Burkholderia cepacia*, *Nannocystis exedens*, *Achromobacter xylosoxidans*	Biocontrol *A. flavus* growth Inhibit *A. parasiticus* growth Inhibit aflatoxin production	[32,33] [34] [35,36]
Fungi	*Aspergillus*	*A. flavus*, *A. parasiticus*, *A. niger*, *A. oryzae*, *A. clavatus*	Inhibit *A. flavus* growth Inhibit several plant pathogens	[37,38,39,40] [41,42]
	*Trichoderma*	*T. harzianum*, *T. viride*, *T. longibrachiatum*,	Biocontrol *A. flavus* growth	[43,44]
	*Penicillium*	*P. chrysogenum*, *P. nalgiovense*	Inhibit aflatoxin production	[45,46]
Yeast	xx	*Debaryomyces hansenii* (marine), *D. hansenii* (native), *Saccharomyces cerevisiae*, *Kluyveromyces* spp., *Pichia anomala*, *Candida maltosa*	Inhibit several common pathogenic fungi Inhibit mycotoxins production	[47,48] [49,50] [51,52]

**Table 2 toxins-12-00024-t002:** Inhibitory compounds produced by antagonists against aflatoxigenic molds.

Antagonists	Inhibitory Compounds	Main Characteristics of the Compounds	References
*Bacillus* spp.	Lipopeptides: surfactin, iturin A and fengycin	Stable after autoclaving	[18,89]
	Bacillomycin D	Completely inhibit *A. flavus* growth	[90]
	Protease	Stable under high alkaline conditions	[15]
	Oligopeptide (L-Asp-L-Orn)	Be able to enter into cells of *A. flavus*	[91]
*P. fluorescens*	Chitinolytic enzyme	Extracellular enzyme	[56]
*Lactobacillus* spp. *Lactobacillus* spp.	Lactic acid	With 60% antifungal activity at 0.02 mg/mL	[22,25]
	Phenyllactic (PLA)	Lose activity after neutralization treatment	
	Hydroxyphenyllactic acid (OH-PLA)	Show strong antifungal ability at the lowest concentration of 1 mg/mL	[92]
	Indole lactic acid (ILA)	About 1 mg/mL was sufficient to inhibit aflatoxins production by 90%	
	2-butyl-4-hexyloctahydro-1H-indene, Oleic acid, palmitic acid, linoleic acid and 2,4-di-tertbutylphenol	In cell-free supernatant; resistant to sterilization and proteolytic enzymes	[22,24]
	Peptides	Completely inhibit *A. flavus* growth on corn	[59]
*Streptomyces* spp.	2-methylisoborneol	A volatile organic compound with ability against storage fungi such as *F. moniliforme* and *A. flavus* in vitro	[30]
	Aflastatin A	Completely inhibit *A. parasiticus* growth at a concentration of 0.5 μg/mL	[93]
	Dioctatin A	Strongly inhibit aflatoxin production	[94]
	Dimethyl trisulfide	Completely control *A. flavus growth*	[95]
	Dimethyl disulfide	Affect mycelial growth and sporulation	[30]
	Benzenamine	Completely inhibit *A. flavus* growth at 1 mL/L	[95]
	Chitinase	With thermal stability and broad pH stability	[29,96]
Yeast strains	2-phenylethanol	Inhibit conidial germination and aflatoxin production	[51]
	Isoamyl acetate	Inhibit the growth of several pathogenic fungi	[52]
	Isoamyl alcohol		
	4-Hydroxyphenethyl alcohol	In cell-free supernatant extract; stable at high temperatures	[97]
	4,4-Dimethyloxazole		
	1,2-Benzenedicarboxylic acid dioctyl ester		
	Chitinase	With ability to cause hyphal lysis and deterioration	[98]
	*β*-1,3-glucanase		
*T. harzianum*	Protease P6281	Stable in pH = 2.5–6.0; with ability to inhibit conidial germination and mycelial growth	[99]
*Serratia marcescens*	Chitinase	With ability to degrade fungal cell walls	[32]
*Penicillium chrysogenum*	Antifungal protein PgAFP	Molecular mass is 6494 Da; belong to small, cysteine-rich, and basic proteins	[84]
*Aspergillus clavatus*	Antifungal peptide	Molecular mass = 5773 Da; with thermostability	[42]
*Achromobacter xylosoxidans*	Cyclo(L-Leucyl-L-Prolyl)	Inhibit aflatoxin production by repressing transcription of aflatoxin-related genes	[36]

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
