# Peer review of "Control of Aflatoxigenic Molds by Antagonistic Microorganisms: Inhibitory Behaviors, Bioactive Compounds, Related Mechanisms, and Influencing Factors"

_toxins, 2020, doi:10.3390/toxins12010024_

Round 1
Reviewer 1 Report
The review comprehensively summarizes the subject.
There are many language errors. Though they do not hinder the comprehension of the text, I feel the present review should be read carefully and amended.
Author Response
Point 1: The review comprehensively summarizes the subject.
There are many language errors. Though they do not hinder the comprehension of the text, I feel the present review should be read carefully and amended.
Response 1: we have checked syntax, grammar, choice of words throughout the manuscript. According to the suggestions, a lot of sentences were also deleted or rewritten. All changes were highlighted in red in the text.
Overall, thank you very much for the positive comments on our articles and the valuable suggestions. Thank you very much.
Reviewer 2 Report
Manuscript ID: Toxins 630450
Authors: Blind Review
Title: Biological control of aflatoxigenic molds by antagonistic microorganisms: inhibitory behaviors, bioactive compounds, related mechanisms and influencing factors
Recommendation: Reject
In this review, the authors focus on the biocontrol strategies limiting the growth and/or aflatoxin production by the main producers of this mycotoxin. They review the types of species with antiaflatoxigenic potential, the range of active compounds they produce, the mechanisms leading to inhibition and, finally, the parameters influencing aflatoxin production.
In my opinion, the topic is of interest for the readers of Toxins but, as it is written, I must reject this manuscript due to the poor level of the written english. There are sections and a lot of sentences that must be rewritten before consideration for publication. Besides that, the authors should bear in mind that species names must be written in italics (check the text and mainly references), the legend of the figures should be self-explaining, and that technical words of this field of research must be used instead of others such as "Destroying the Cell-wall structure" (in Figure 4). Taking everything into consideration, my decision is Reject. I include a copy of the text with all the annotations I have done. It may be helpful for the improvement of the text.

Author Response
Point 1: In my opinion, the topic is of interest for the readers of Toxins but, as it is written, I must reject this manuscript due to the poor level of the written English. There are sections and a lot of sentences that must be rewritten before consideration for publication. Besides that, the authors should bear in mind that species names must be written in italics (check the text and mainly references), the legend of the figures should be self-explaining, and that technical words of this field of research must be used instead of others such as "Destroying the Cell-wall structure" (in Figure 4). Taking everything into consideration, my decision is Reject. I include a copy of the text with all the annotations I have done. It may be helpful for the improvement of the text
Response 1: According to the suggestions, we have checked the text and mainly references to ensure species names were written in italics.
About the written English in the text, we have checked syntax, grammar, choice of words throughout the manuscript. According to the suggestions, a lot of sentences were also deleted or rewritten.
Overall, we sincerely thanks for the professional reviews on our article and the help of language corrections. Thank you again for the valuable suggestions to improve the quality of our manuscript. Thank you very much.
Reviewer 3 Report
The article presented on biological control of toxigenic fungi that produce aflatoxins is of interest, as it is a novel compilation about a family of mycotoxins of high toxicological interest, and deserves to be published in Toxins.
However, some considerations must be made:
1.- The title must be modified because it is repetitive. I recommend that the word "Biological" be deleted or "by antogonistic microorganisms" be deleted, at the choice of the authors.
2.- Figure 3 is superfluous and should be removed, as well as any mention of it that appears in the text, such as the paragraph included in lines 261-265 "The corresponding ... Figure 3".
3.- On line 74 you write "Bucillus", and you must say "Bacillus"
4.- In Table 1, "Antifungal activity" should be changed to "Activity", since in the column reference is made to both, fungi and mycotoxins.
5.- Bibliographic citations 38 and 108 are repeated, also correct in the text. Check that there are no other errors in the bibliography.
6.- As a keyword, "Food safety" should be deleted since the article is not interested in contaminated food.
7.- In the introduction, the paragraph between lines 34 and 48, whose objective is to show the importance of food contamination, should be eliminated, since the article does not address food control.
Author Response
Point 1: The title must be modified because it is repetitive. I recommend that the word "Biological" be deleted or "by antogonistic microorganisms" be deleted, at the choice of the authors.
Response 1: The word "Biological" was deleted (Line 2).
Point 2: Figure 3 is superfluous and should be removed, as well as any mention of it that appears in the text, such as the paragraph included in lines 261-265 "The corresponding ... Figure 3".
Response 2: Figure 3 was removed finally.
Point 3: On line 74 you write, and you must say "Bacillus"
Response 3: The word "Bucillus" has been corrected as "Bacillus".
Point 4: In Table 1, "Antifungal activity" should be changed to "Activity", since in the column reference is made to both, fungi and mycotoxins.
Response 4: "Antifungal activity" has been changed to "Activity" (Line 64, Table 1).
Point 5: Bibliographic citations 38 and 108 are repeated, also correct in the text. Check that there are no other errors in the bibliography.
Response 5: It was corrected and we have checked the text to avoid the similar problems.
Point 6: As a keyword, "Food safety" should be deleted since the article is not interested in contaminated food.
Response 6: "Food safety" was deleted (Line 19).
Point 7: In the introduction, the paragraph between lines 34 and 48, whose objective is to show the importance of food contamination, should be eliminated, since the article does not address food control.
Response 7: After a serious discussion with the authors, we eliminated the paragraph finally.
Overall, thank you for the professional review work on our article! Thanks for the positive reviews on our articles. Thank you very much.
Round 2
Reviewer 2 Report
Although the authors have improved the quality of the text, it still needs extensive review and editing. Attached to this report, I add a pdf file with some of the multiple errors that still remain in the second version of the manuscript. I think it will help the authors. However, they should carefully review the whole manuscript.
That is why my decision is Major Review.

Author Response
Point 1: Although the authors have improved the quality of the text, it still needs extensive review and editing. Attached to this report, I add a pdf file with some of the multiple errors that still remain in the second version of the manuscript. I think it will help the authors. However, they should carefully review the whole manuscript.
Response 1: According to the suggestions, we carefully revised the manuscript again. Some errors, the choice of words and the use of Articles were especially checked and corrected. All changes in the manuscript were highlighted using red colored text.
Thank you again. Thanks very much for the valuable suggestions to improve the quality of our manuscript.
Round 3
Reviewer 2 Report
The authors have improved the manuscript significantly with the corrections done to the quality of the written english. However, some errors remain. See for example in line 159 ("filed" instead of "field"), line 166 ("secret" instead of "secrete", line 28 (use "a loss of nutritional value"). I recommend the acceptance of the manuscript once these minor corrections are done.
Author Response
Point 1: The authors have improved the manuscript significantly with the corrections done to the quality of the written english. However, some errors remain. See for example in line 159 ("filed" instead of "field"), line 166 ("secret" instead of "secrete", line 28 (use "a loss of nutritional value"). I recommend the acceptance of the manuscript once these minor corrections are done.
Response 1: According to the suggestions, the errors were corrected (Line 28, 160 and 167). Additionally, some other errors such as “A. paraciticus” instead of “A. parasiticus” (Table 1), “Aspergillius” instead of “Aspergillus” (Line 255 and 360) have also been corrected.
The other changes in the manuscript were highlighted using red colored text.
Thank you very much again. Thanks for the patience and help to improve the quality of our manuscript. Thank you.